# Monitoring Genetic Erosion of Aromatic and Medicinal Plant Species in Alentejo (South Portugal)

**DOI:** 10.3390/plants12142588

**Published:** 2023-07-08

**Authors:** Orlanda Póvoa, Violeta Lopes, Ana Maria Barata, Noémia Farinha

**Affiliations:** 1VALORIZA—Centro de Investigação para a Valorização de Recursos Endógenos, Instituto Politécnico de Portalegre, Praça do Município 11, 7300-110 Portalegre, Portugal; 2Instituto Politécnico de Portalegre, Praça do Município 11, 7300-110 Portalegre, Portugal; nfarinha@ipportalegre.pt; 3Banco Português de Germoplasma Vegetal (BPGV), Instituto Nacional de Investigação Agrária e Veterinária, Quinta de S. José, S. Pedro de Merelim, 4700-859 Braga, Portugal; violeta.lopes@iniav.pt (V.L.); anamaria.barata@iniav.pt (A.M.B.)

**Keywords:** ethnobotany, genetic diversity, germplasm collection, plant populations, sampling, traditional ecological knowledge

## Abstract

The main goal of this work was to study the genetic erosion risk of plants with aromatic, medicinal and gastronomic applications in Portugal, particularly in the Alentejo region. The target species were coriander (*Coriandrum sativum* L.), hart’s pennyroyal (*Mentha cervina* L.) and pennyroyal (*Mentha pulegium* L.). The methodology involved direct observations and surveys (2002/2003 and 2011). The GE formula applied in Hammer’s studies was used to estimate genetic erosion. The main factors causing genetic erosion were the primary drivers of biodiversity loss: habitat loss, invasive species, and overexploitation influenced by human intervention such as the clearing of watercourses, vegetation control, grazing and desertification. The results indicate a reduction in individuals per species in Alentejo, with a net erosion loss of 11% for *M. pulegium*, 32% for *M. cervina* and 33% for *C. sativum*. The overall loss of accessions (genetic erosion risk) was higher in cultivated accessions (33%) than in wild accessions (11%), with an annual genetic erosion rate of 3.7% and 1.2%, respectively. The annual risk of genetic erosion for *M. pulegium* accessions collected in a natural habitat was 0.6%, which is much lower than the 3.7% for *M. cervina*. These results consolidate the importance of collecting and conserving genetic resources.

## 1. Introduction

Genomic erosion is a pervasive—but frequently overlooked—consequence of the many threats faced by wild populations, such as overexploitation, invasive species, emerging infectious, diseases, hybridization, and habitat and environmental changes [1].

Genetic erosion can be defined as the “loss of genetic diversity, in a particular location and over a particular period, including the loss of individual genes, and the loss of particular combinations of genes such as those manifested in landraces or varieties. It is thus a function of change of genetic diversity over time” [2]. Estimating past genetic erosion [3] provided a useful list of features or indicators that could be measured singly or in combination on individuals and populations of a given species in a defined area as part of a systematic effort to monitor changes in genetic diversity in the species. In genetic erosion studies involving direct temporal comparisons, the same population(s) are studied at different times. This involves re-sampling and the direct comparison of samples collected on different occasions and conserved ex situ in gene banks or as dried plant material in a herbarium. The basis of the comparison could be local knowledge, conventional morphological characterization, agronomic evaluation, or molecular markers. Estimation of change in genetic diversity over time can be carried out by comparing the original and re-sampled population data, either crudely in terms of the presence or absence of species, or more objectively by comparing numbers of individuals present for the species previously found, or even more precisely by comparing allele frequencies, genetic diversity, and levels of inbreeding in the original with the current accessions collected from the same site. It is possible to assess genetic diversity changes for the selected populations in the intervening period between the two or more collections. Identification of the drivers of genetic diversity change: while revisiting the sites, detailed socio-economic data should be collected from farmers and other stakeholders in the local area to enable an assessment to be made of the socio-economic drivers of genetic diversity change [4,5,6,7].

Genetic erosion, according to the European Environment Agency (CHM Biodiversity), is the loss of genetic diversity between and within populations of the same species over time, or the reduction of the genetic basis of a species due to human intervention, environmental changes, etc.

Habitat destruction, degradation, fragmentation or conversion for agriculture, ranching, horticulture, mining, ecotourism, industry, population fragmentation, commercial over-harvesting to satisfy urban and export demands, and overgrazing are some of the human exerted pressures [8] on native populations, their biology and their potential to respond to environmental shifts leading to dwindling population sizes of plants (for many MAPs, population size is directly related to genetic diversity), population densities, diminished fitness, enhanced isolation, genetic erosion, and species extinction. Population fragmentation, isolation, and decreased population densities/sizes force inbreeding within sites, modifying patterns of gene exchange, and pollen and seed movement between fragmented populations leading to genetic erosion. For wild-collected MAPs the impact of over-harvesting depends on the plant part collected, biology, range, distribution, and economic value. Populations may disappear more rapidly due to over collection than from fragmentation or habitat destruction [9].

As genetic erosion research has evolved, three main measurement targets have emerged. The quantification may be direct, or through proxies such as numbers of farmers or villages. These crop diversity analyses have been conducted at a wide range of geographic scales, from local (e.g., farm, population or genebank accession), to community and agroecological landscapes, to country, regional and global scales. Intermediate time frame studies often compile and report diversity changes at standardized intervals, such as per decade. As with other parameters, the sources of information used to document change in crop diversity also vary widely and may be used in combination. Direct field observations provided the first lines of evidence for genetic erosion. Local knowledge, gathered through interviews with farmers and their families, community meetings, and surveys, have been widely used to assess change and document farmers’ perspectives [10,11,12,13,14,15,16].

There is increasing evidence for the significant value and potential of medicinal and aromatic plants (MAPs) worldwide. Among other non-wood forest products, MAPs are considered a key element of sustainable forest management and economic development. As part of Mediterranean cultural heritage, these plants are a major driver of rural tourism and in many areas represent an important raw material for various bio-based industrial sectors. In addition to their economic value, MAPs enhance social integration and maintain gender balance as harvesting and processing MAPs is clearly a female-dominated task [17]. Despite the prominent contribution of MAPs to local development, conservation of biodiversity and the development of the traditional Mediterranean food system, many challenges and knowledge gaps could potentially place the sector’s development at risk. In Portugal, there are good market prospects for an increase in production, contributing to the development of rural populations, promoting farmers’ well-being and fixing populations in the rural area [18,19].

As the knowledge about traditional use of plants is disappearing, ethnobotany has an important role in saving the remaining information [20]. The farmers’ knowledge on the use and production of landraces and local spontaneous aromatic and medicinal plants is important for the conservation of regional plant genetic resources.

Mint is the common name for most species of *Mentha* (Lamiaceae). In Portugal, *Mentha pulegium* L. is mostly known as “poejo” (pennyroyal). Pennyroyal is a very aromatic, herbaceous species, frequent in wet meadows and pastures, in ravines, banks and water line’s dry beds, ponds, lagoons, and other temporarily flooded or flooded sites, which are sometimes nitrified. Pennyroyal has a preference for acidic soils, with permanent or seasonal edaphic humidity in mainland Portugal and on the Azores and Madeira archipelagos.

*M. cervina* (hart’s pennyroyal), is also known in Portugal as “poejo-fino” (thin-pennyroyal), among other common names. It is a rare species from the Natura habitat 3130 pt5 “temporary deep ponds” [21]. It appears in the ecological zones of temporary ponds, beds of watercourses that dry up in summer, river flood beds, and in sandy or stony places. Usually, both pennyroyal and hart’s pennyroyal have the same seasoning and medicinal uses [22]. Hart’s pennyroyal is widely used in Alentejo and in the Upper Douro region in the Sabor river basin as a condiment for river fish dishes.

Pennyroyal is traditionally used in culinary decoration, in the well-known pennyroyal liquor, and as a culinary herb, primarily in the preparation of piso paste, and typical fish (caldeirada) and bread (açorda and migas) Portuguese dishes. The infusion and/or pennyroyal liquor is widely used as a digestive and/or to tackle colds. In Portugal, the medicinal use of *Mentha* ssp. is similiar, being mainly related to respiratory and digestive disorders.

Coriander (*Coriandrum sativum* L., Apiaceae) leaves are widely used in the traditional and ancient recipes of various regions of Portugal, especially in Alentejo in its popular açorda and fish soup (shark). It is an annual plant of the Apiaceae family, very aromatic, with finely cut upper leaves and small white or pink flowers. Its very round brown seeds are very popular in cooking, giving a distinctive aromatic flavor. It is probably native to the Mediterranean basin where the Greeks and Romans used it in the preparation of dishes and drinks. In the Middle Ages, it was cultivated in monastery gardens. Its seeds and leaves are widely used in Indian and Arabian cuisine. In Portugal, the seeds are also used to make confetti and other sweets. The coriander seeds and leaves are widely used to season meat and vegetables. The essential oil is used in soaps, deodorants and toothpaste. With digestive, antiseptic, diuretic and soothing properties, it is used as a therapeutic resource in situations of high cholesterol, colic, conjunctivitis, diarrhea, toothache, migraine, fever, swellings, indigestion, neuralgia, rheumatism and measles [23].

Although it is generally accepted that a significant amount of genetic erosion has occurred and is occurring in Portugal [24,25,26,27], there are limited data on its degree and extent. In general, the last decade has seen serious genetic erosion in several Portuguese habitats due to the pressure of urbanization, tourist developments and golf courses. The introduction of invasive species has also greatly contributed to genetic erosion. Concerning the landraces, genetic erosion is due to the replacement of traditional cultivars with commercial ones. In 2004, the “Networking on Conservation and Use of Medicinal, Aromatic and Culinary Plants Genetic Resources in Portugal” was established in Portugal [28]. The network aimed to undertake surveys, to perform systematic collecting missions, to collect anthropological information on the aromatic and medicinal species existing in different regions of the country, and to conserve and evaluate the collected germplasm. A total of 1394 ethnobotanical questionnaires were elaborated at the national level based on 804 interviews. In addition, the network also aims to contribute to overtaking the problem of arbitrary harvesting of wild plants, which is a severe factor in genetic erosion [29].

In Portugal, studies on genetic erosion have focused on indigenous animal breeds. In plants, as far as grapevines are concerned, the Portuguese varieties were observed from the point of view of grapevine diversity [24,25]. Concerning cereal crops, genetic erosion analyses were performed in ex situ collections of genetic resources [26]. Rocha et al. [27] analyzed genetic erosion assessments between ex situ collections in the Portuguese Genebank (BPGV) and the re-sampling in the same territory for common bean, rye and maize germplasm. The target territory was an area where a genetic reserve area of the Peneda Gerês National Park (PNPG) was included. One of the main results of the study was the tendency for the decrease in diversity to be associated with the ageing of residents, and the consequent abandonment of agricultural activities. Farmers grew only enough for their own consumption. In addition, there was the progressive abandonment of traditional varieties and the increasing use of improved commercial varieties, particularly of maize, with the consequent disappearance of traditional farming systems and traditional varieties [27,30,31].

In spite of the evident declines in crop diversity that have been raised, the magnitude, trajectory, drivers and significance of these losses remain insufficiently understood [15].

The present study aimed to estimate the extent of genetic erosion in Alentejo (Southern Portugal) of three taxa of aromatic, medicinal and condiment group species, *M. pulegium, M. cervina* and *Coriandrum sativum.*

In 2002–2003, a survey on traditional ecological knowledge was carried out to assess the causes of genetic erosion of the studied taxa—coriander (*Coriandrum sativum* L.), hart’s pennyroyal (*Mentha cervina* L.) and pennyroyal (*Mentha pulegium* L.). The concept of crop genetic erosion, as applied in this research, is the loss of crop diversity in a defined area over a period of time, typically measured by the decline of species, varieties and/or within varieties [15,32]. In 2011, the same wild (*n* = 28) and cultivated (*n* = 111) populations were studied, using indirect data (loss of accessions) and field observations to estimate the risk of genetic erosion and its causes.

## 2. Results

### 2.1. Cultivated Accessions

A total of 95 accessions were considered (28 *M. cervina*, 19 *M. pulegium*, 48 *C. sativum*). The genetic erosion risk estimated for cultivated accessions was similar among the three studied species (32–33%) (Figure 1). Accessions loss was mainly due to the informant’s death (67%), migration or abandonment of farming (due to aging) (Figure 2). Most of the informants in our surveys were women (52%) [33]. Other people lost seeds or plants by different causes such as: drought, pests, etc. The average time of disappearance of accessions was around 8 years. It is assumed that the Alentejo tradition of inheritance, exchange and offering of seeds and cuttings between neighbors had a protective effect and delayed the disappearance of accessions.

### 2.2. Wild Accessions

The main causes of genetic erosion identified by the informants in the 2002–2003 surveys were: desertification, soil erosion, excessive collection from natural habitats, overgrazing, shrub invasion of natural habitats, riparian vegetation cleaning, indiscriminate herbicide application and agricultural machinery use.

Over-recollection from natural habitats and overgrazing were the main causes identified for both *Mentha* species. *M. cervina* was considered more endangered by the informants and more causal factors for genetic erosion were identified for this taxon (Figure 3).

A total of 28 accessions (6 *M. cervina*, 22 *M. pulegium*) were considered for genetic erosion assessment in the 2011 field surveys; the majority (57%) maintained their relative abundance in the natural habitat. *M. cervina* had a lower relative abundance in 25% of the places, whereas *M. pulegium* augmented its relative abundance in 20% of the accession’s sites, some of which had a high grazing pressure (Figure 4). Only habitat destruction, vegetation invasion, mechanization/herbicides application and grazing were observed; however, stream cleaning is difficult to distinguish from habitat destruction, and excessive re-collection and drought were not evaluated due to the lack of comparative methodology. *M. cervina* had the highest genetic erosion risk (33%) (Figure 5) due mainly to habitat destruction (e.g., construction of river dams). *M. pulegium* disappeared in some places (5%) mainly due to mechanization and herbicide application. Seeds were dispersed by wind and natural vegetative cuttings were also dispersed by streams, as [34] associated, which, when joined with less unfavorable local climatic conditions, contributes to a lower risk of genetic erosion. Although grazing is generally considered a threat for wild *Mentha* spp. conservation, present results indicated that a relative abundance persisted after grazing.

## 3. Discussion

The relationship between population size and loss of genetic diversity has been well established and quantified. Generally, smaller populations tend to lose genetic variation by genetic drift much more quickly than larger populations [35,36,37] and the shorter the generation length (that is, time to reproductive maturity), the more rapid the diversity loss in absolute time [35,38].

This relationship has also been examined at the species level, and various reviews have found that restricted or rare species are generally less genetically diverse than more common plant species [35]; although, some rare species have a higher genetic diversity [39]. Considering a sexually reproducing diploid species that is mainly an out breeder, mating among relatives (inbreeding) is more likely in smaller populations. Also, the process is cumulative, so that, over time, mating between unrelated individuals becomes impossible. Increased homozygosity also leads to reduced reproduction and survival (i.e., lower reproductive fitness) and ultimately a higher risk of extinction [35,40]. This cascade of events that results from increased inbreeding is termed “inbreeding depression” [35]. For species that have lost large amounts of habitat and census numbers, it would be expected that considerable genetic diversity would also have been lost [35,41]. This can be particularly serious for self-incompatible species [35].

Overall, accession loss (genetic erosion risk) was higher in cultivated accessions (33%) than in wild accessions (11%) (Figure 6), with annual genetic erosion rates of 3.7% and 1.2%, respectively. Genetic erosion risk for the *C. sativum* accessions collected in 2002 was 32% (3.6% annual). Overall, genetic erosion risk from the accessions collected before 2002 was 38.9%. All *C. sativum* accessions from the 1991 to 1992 collections disappeared or were not found during resampling in 2011. Regarding the collection in 1996–1997, the genetic erosion risk was 20% (1.4% annual), and for the collection in 1999–2000, the calculated genetic erosion risk was 36.4% (3.0% annual). The annual genetic erosion risk for the *M. pulegium* accessions collected in the natural habitat in 2002 was 0.6%, which is much lower than the 3.7% for *M. cervina*. In Albania (1941 to 1993), the genetic erosion on cereal landraces was much higher than in the present results, representing 72.4% at the regional level and 79.2% at the country level; similarly, in Italy (1950 to 1983/86), vegetables showed an average genetic erosion of 82%, but *Ocimum basilicum*, a locally popular aromatic plant, only had a 33% genetic erosion rate [8], similar to the rates found in the present study. Hammer et al. [10] estimated the annual genetic erosion of field crops as being 2.45% in Albania and 3.88% in Italy. These values are approximate to our results for cultivated accessions. As in other studies [17], most of the informants in our surveys were women (52%) as harvesting and processing MAPs is a female-dominated task.

Teklu and Hammer [12] estimated wheat landraces genetic erosion in Ethiopia by resampling, obtaining rates above 77.8%. The replacement of landraces by other crops with a higher economic value for marketing was the main factor for the end of landrace cultivation. From the results of this study, it appears that for coriander, the main force for genetic erosion was crop abandonment. Similarly, genetic erosion in mint was due to the impact of biotic and abiotic factors.

Outcrossing is considered the best precondition for resisting genetic erosion since the whole range of genes will be naturally spread throughout the population [10], and therefore this may explain the *Mentha pulegium* resistance, whereas the vegetative propagated *M. cervina* had the highest rates of genetic erosion risk. The seed dispersion method and broader potential habitat of *Mentha pulegium* contributed to lower wild accession loss, while habitat specificity (Natura habitat 3130 pt5 ‘temporary deep ponds), lack of seed production and viability observed in *M. cervina* made this taxon especially sensitive to habitat destruction. In Portugal, like worldwide [42], riparian habitats, which are situated along rivers in very close proximity of settlements where local people use its resources in the highest degree, are especially threatened.

The data collected also underline the need to conserve this considerable biodiversity, using farm conservation methods, which would allow for a greater adaptation and evolution of traditional varieties in favorable conditions of the Mediterranean basin [43]. Sustainable management agriculture systems could contribute to biodiversity conservation through the maintenance of wild populations growing as weeds in cultivated fields [42].

Traditions supported by isolation in mountainous regions and a low living standard preserved at least a part of the traditional crop landraces in Albania and Italy [10]. Traditions as neighbors’ share of plants and seeds and family heritage also contribute to saving coriander and *Mentha* spp. accessions in our study. As [44] have considered, the loss of varieties does not necessarily lead to the erosion of the genetic diversity of the crop or the reduction of diversity in a region. Farmers have always exchanged seeds between different regions and selected promising variants.

The main reason for the genetic erosion of cultivated plants in the Alentejo region (Portugal) is the human depopulation of the region, with the rural population aging and abandonment of traditional farming, and the consequent loss of cultivated regional plant genetic resources.

## 4. Materials and Methods

### 4.1. Target Area of Study

Alentejo is a region of 31 605 km^2^ located in southern Portugal with the Tagus river (39°39′49″) as its northern border, and the Algarve region with the Monchique and Caldeirão mountains (37°19′08″) as its southern border. Spain (06°55′53″) is at the eastern border and the Atlantic Ocean (−09°00′16″) is at the western border. Its longest north–south distance is 260 km and the longest east–west distance is 181 km. Its altitude varies from zero at ocean level to 1027 m in northern Alentejo in the São Mamede Mountain [45]. The region is usually divided into 4 sub-regions: northern Alentejo, central Alentejo, southern Alentejo and littoral Alentejo. In addition to the Tagus river at its northern border, the region also has the Guadiana river near the Spanish border, and the Sado and Mira rivers in its central and littoral sub-regions.

Its climate is typical of the Mediterranean (Table 1), sub-humid dry and mesothermal, with warm dry summers and cold rainy winters. The lowest average temperature in January (the coldest month) is 4.1 °C and the highest average temperature in August (the warmest month) is 36 °C. The annual accumulated rainfall is 817 mm in southern Alentejo and 947 mm in the north of Alentejo, with 243 days without any rainfall. North Alentejo is slightly colder and wetter, and southern Alentejo is slightly warmer and drier [45]. The region has one of the lowest population densities in Portugal (Table 1) and in the European Union, with a rate of 115.4 inhabitants/km^2^ in the country and only 14.5 inhabitants/km^2^ in southern Alentejo, with negative growth and people-aging tendencies [45]. Nowadays, [46] the regional population loss continues, with a 7% population loss (2011–2021).

People’s employment depends mainly on: 1—wholesale and retail trade, 2—agriculture, livestock production and the forestry sector, and 3—manufacturing. The average farm area (51 ha) is the highest in Portugal (12 ha), being even higher in South Alentejo (66 ha) [45].

The typical agriculture is multifunctional with cork oak agroforestry systems and permanent pastures for extensive livestock grazing (1,205,919 ha). Livestock consists of cows (cattle and milk), pigs, sheep and goats. Horses are in the minority but are regionally important for cultural reasons. Cereals, olives and vineyards are also important regional crops. Family horticulture, where medicinal and aromatic plants are cultivated, is a minor activity in area (1593 ha), but important for supporting livelihoods and biodiversity conservation [45]. In Alentejo, individual producers with medium or higher qualifications represented an indicator of 14% and in the country it was less than 10%; individual producers working full-time on the farm was 12% and in the country it was around 22%; female individual producers in the Alentejo had an indicator of 22% and in the country the indicator was 30%.

### 4.2. Selection of Study Sites and Farmers

#### 4.2.1. Sample Collection

Thirty-four accessions of coriander were previously collected within a joint mission from our team and the Portuguese Gene Bank team (BPGV): 11 in 1992; 9 in 1997; and 15 in 2000. Only seeds from local proveniences (landraces) were collected for coriander. Passport data were registered. *Mentha* spp. potential sites were also gathered from herbarium vouchers; the accessions collection took place in home gardens and natural habitats. During project Agro 34 field missions, in 2002 and 2003, 30 accessions of coriander, 34 accessions of *M. cervina* and 31 accessions of *M. pulegium* were collected; the location of the collection sites was identified to better represent the Alentejo region, covering an area of 31,551 km^2^, representing 33% of mainland Portugal. Seed samples of *M. pulegium* and *C. sativum* and vegetative samples of *M. cervina* were also collected in 2002–2003 and sent to the Portuguese Gene Bank for ex situ conservation. 

At each site, accession passport data, soil samples and ethno-botanical data were collected. Each site was recorded with a GPS device and then downloaded to GIS (Geographic Information System) software to produce the interview location map.

#### 4.2.2. Resampling (2011)

Accession passport files and queries from the previous 2002–2003 collection missions were used to track informant’s contacts and wild habitat location sites (Appendix A).

The incomplete database of 34 coriander accessions harvested before 2002 made it difficult to trace them. However, it allowed 15 of them to be traced. Nevertheless, they were not considered for estimating the risk of genetic erosion. This problem was even more severe for the 11 accessions collected in 1992, of which only two sites were found. Van de Wouw et al. [44] considered the possible problem of copying the route and locations of the original collection mission.

#### 4.2.3. Data Collection and Analysis

A total of 123 accessions were considered: 28 cultivated and 6 wild of *M. cervina*; 19 cultivated and 22 wild of *M. pulegium;* and 48 cultivated of *C. sativum* (Figure 7).

Tracking down missing informants was particularly difficult. In the best circumstances, neighbors told us if the person had died or changed address. In these cases, attempts were made to find out if the seeds or plants were left as an inheritance or even, as a neighbor’s gift.

For wild accessions of both *Mentha* species, in addition to the identification of the original collection sites in nature, evidence of the risk of genetic erosion was also observed.

Brown [3] states that the essential statistic for estimating genetic erosion is the proportion of variants (alleles, genotypes or populations) lost or likely to be lost in a given time period (for example, a decade), specifying the sample basis that is the subject of the inferences.

The genetic erosion risk was estimated by the percentage of accessions lost. A similar method (Genetic erosion (GE) = 100%—GI (Genetic integrity)) was used by Hammer [10,11] for calculating genetic erosion of several Italian and Albanian landraces. Resampling was also used by [12] to estimate wheat landraces genetic erosion in Ethiopia: E = 100%—GI, where GE is genetic erosion and GI is genetic integrity, which is given as: GI = N2/N1 × 100. The number of farmers cultivating landraces and their relative area coverage were determined. The number of landraces cultivated recently as compared to the previous number was the basis for the calculation of genetic erosion [15,47,48]. A reduction in richness (that is a reduction in the total number of crops, varieties or alleles) is a better indicator for genetic erosion, as it does recognize the dynamics in the system [12].

## 5. Conclusions

Most threats to biodiversity result from human actions, expressed in the overuse of natural resources for fuel, fodder, manure, grazing and collecting of plants in their natural habitats for ornamental or medicinal purposes. This overuse leads to the loss of genetic diversity including crop wild relatives. According to the FAO’s State of the World’s Plant Genetic Resources for Food and Agriculture [49], the main cause of genetic erosion for landraces in developed countries is the replacement of local varieties by modern varieties. However, modern cultivars failed to meet the different environmental requirements, cultural preferences and agricultural practices [44]. The main problem in Alentejo for coriander was farm abandonment, not the replacement of varieties, as the Alentejo region is considered an area of low population density. Other genetic erosion causes include the emergence of new pests, weeds and diseases, environmental degradation, urbanization, modern agricultural practices, and land clearing such as deforestation and fires [11,12,44,50]. Another problem identified in Alentejo was overcollection. From these results, ethnobotanical surveys can be useful for assessing genetic erosion risk causes of medicinal and aromatic plants. 

Overall, accessions loss (genetic erosion risk) was higher in cultivated accessions (33%) than in wild accessions (11%) due to the human depopulation of the Alentejo region. The elderly rural population and the abandonment of traditional farming systems caused the loss of cultivated regional plant genetic resources.

*Mentha cervina* in wild habitats had the highest genetic erosion risk (25%) mainly due to habitat destruction. Grazing is usually considered a threat for wild accessions. In grazed areas, the biomass of *Mentha* spp. was highly reduced, but the plants persisted, apparently indicating an adaptation to the grazing effect. However, overgrazing can be a serious genetic erosion risk.

So, complementary strategies of ex situ and in situ conservation should be implemented for the conservation of this rare species (*M. cervina*) because apparently only four wild populations persist in the Alentejo region.

Overall, the high risk of genetic erosion justifies further regional plant genetic resources collection, especially of cultivated plants, being necessary to employ more thorough sampling strategies in future collection programs [8,51]. Further studies with the collected material are also fully recommended [52,53].

## Figures and Tables

**Figure 1 plants-12-02588-f001:**
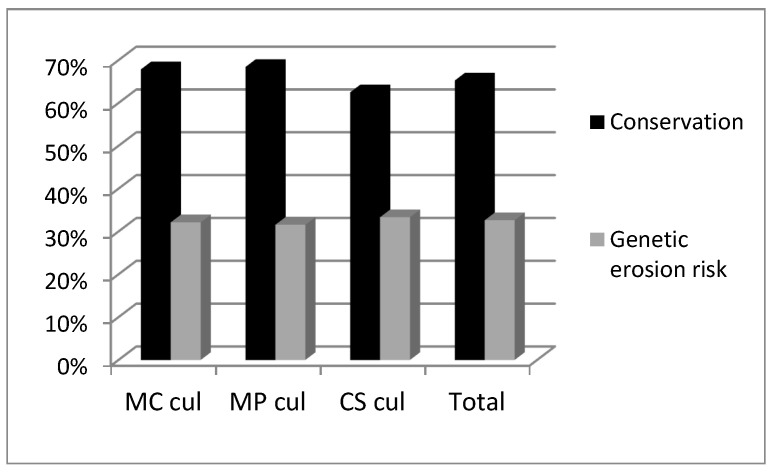
Conservation and risk of genetic erosion for cultivated accessions of *M. cervina* (MC), *M. pulegium* (MP) and *C. sativum* (CS) determined in the period 2002–2011.

**Figure 2 plants-12-02588-f002:**
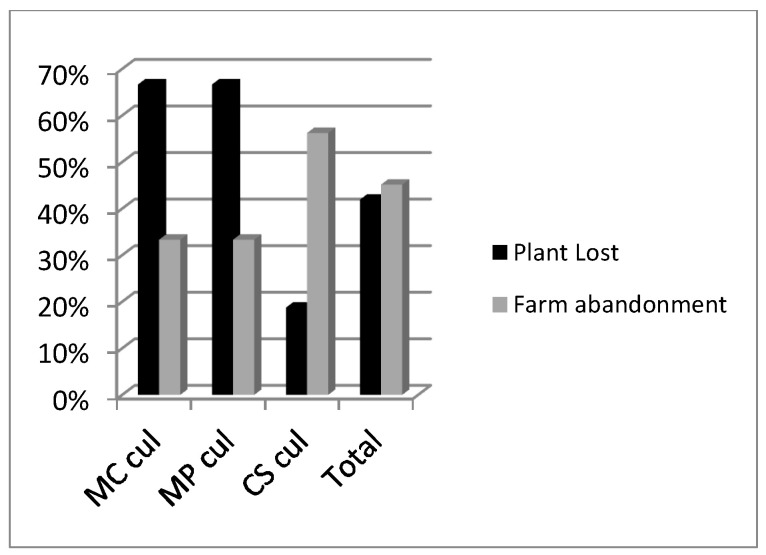
Main causes of genetic erosion risk for *M. cervina* (MC), *M. pulegium* (MP) and *C. sativum* (CS) cultivated (cul) accessions determined in the period 2002–2011.

**Figure 3 plants-12-02588-f003:**
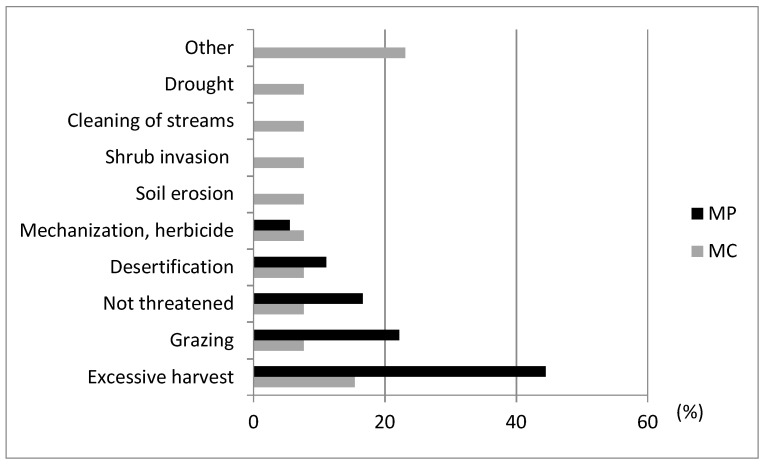
Conservation threats for *Mentha pulegium* (MP) and *M. cervina* (MC) identified by informants in the 2002–2003 ethnobotanical surveys.

**Figure 4 plants-12-02588-f004:**
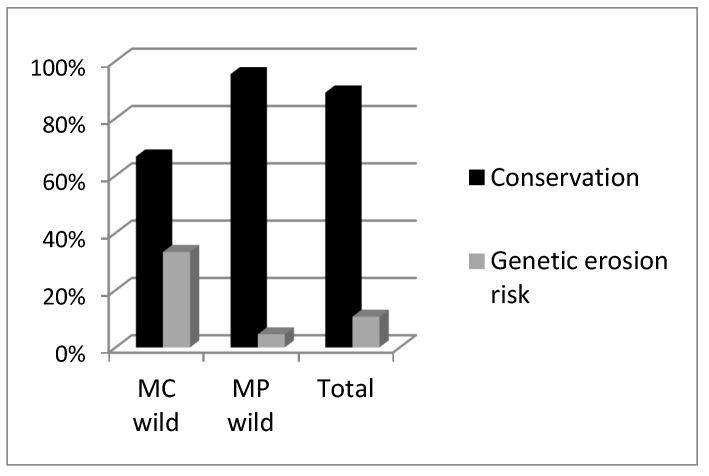
Conservation and genetic erosion risk of *Mentha cervina* (MC) and *M. pulegium* (MP) wild accessions (2002–2011).

**Figure 5 plants-12-02588-f005:**
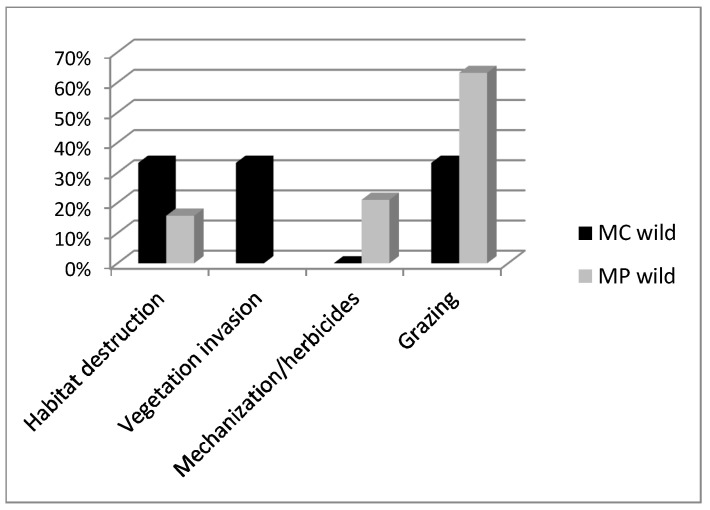
Main conservation threats for *Mentha cervina* (MC) and *M. pulegium* (MP) wild accessions identified during 2011 surveys (period 2002–2011).

**Figure 6 plants-12-02588-f006:**
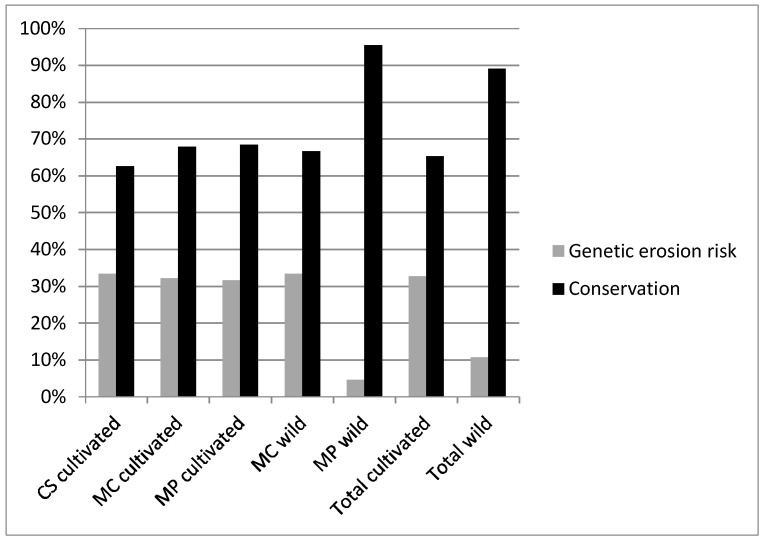
Total genetic erosion risk of *M. cervina* (MC), *M. pulegium* (MP) and *C. sativum* (CS) determined in the period 2002–2011.

**Figure 7 plants-12-02588-f007:**
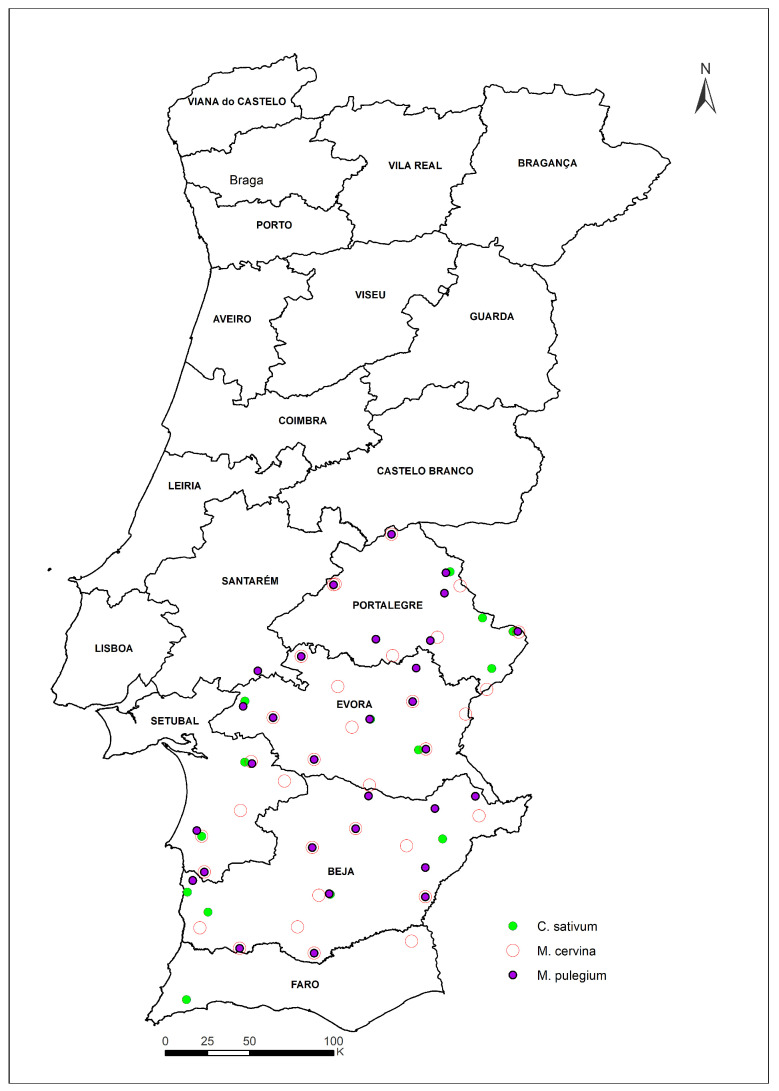
Accession sites in Alentejo region (note: some points overlap).

**Table 1 plants-12-02588-t001:** Climate and population size of the Alentejo region.

Region	Average Temperature (°C)	Annual Precipitation (mm)	Population Size 2011 (Inhabitants/km^2^)	Population Size 2021 (Inhabitants/km^2^)
Alentejo			23.7	22
North Alentejo	16	947	18.2	17
Central Alentejo	17	852	23.1	21
South Alentejo	17	817	14.5	13
Littoral Alentejo			17.8	18

## Data Availability

Not applicable.

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
