# Peer review of "Monitoring Genetic Erosion of Aromatic and Medicinal Plant Species in Alentejo (South Portugal)"

_plants, 2023, doi:10.3390/plants12142588_

Round 1
Reviewer 1 Report
Interesting paper, that highlights the "genetic erosion" that is occurring in many species throughout Portugal. I think the organization of the species being study get a little confusing with line 51 mentioning Mentha pulegium L, and then jumps down to introduce M. cervina (L). I think you could save the information of on M. cervina, and tie it into the paragraph that beings at 58 to maintain the flow of those three plants individually.
Line 97 should use focused instead of focusing.
I wonder if you should bring lines 83 and below that discuss genetic erosion, should be early before the introduction to the species you plan on studying.
Minor errors here and there.
Author Response
"Please see the attachment.

Reviewer 2 Report
This is a well sound ethnobotanical study. Please consider including the following suggestions.
Authors state that “Besides their economic value, MAPs enhance social integration and maintain gender balance as harvesting and processing MAPs is clearly a female dominated task” but there is no gender consideration in the questionnaires or in the discussion section. Please consider revisiting this statement in your discussion section under the light of your results.
L83 Please clarify what do you mean by “genetic erosion”? and please include a reference. Perhaps you can move here L111-117?
L97 change “have focusing”, to “have focused”
L110 needs a reference
L171-172 change “was carried out with a view to assessing” to “was carried out to assess” and “of the taxa studied” to “of the studied taxa”-
L174 add “of time” after “in a defined area over a period”
L176 delete “also”
L208-223 these paragraphs need more references, also do they belong to the Results’ section?
L275-276 Rephrase for clarification “Hammer et al. [22], obtaining 2.45% in Albania and 3.88% in Italy, estimated the annual genetic erosion of field crops.”
Minor editing of english language required.
